**Subject Category:**
Biology (whole organism)

palaeontology/evolution

Osteichthyes, development, dentition, ancestral, micro-CT

**Authors for correspondence:**
Mark Doeland
e-mail: markdoeland@outlook.com
Martin Rücklin
e-mail: martin.rucklin@naturalis.nl

# Tooth replacement in early sarcopterygians

Mark Doeland[1,2], Aidan M. C. Couzens[1],
Philip C. J. Donoghue[3] and Martin Rücklin[1]

[1]Naturalis Biodiversity Center, Postbus 9517, 2300 RA Leiden, The Netherlands
[2]Institute of Biology, Universiteit Leiden, Silviusweg 72, 2333 BE Leiden, The Netherlands
[3]School of Earth Sciences, University of Bristol, Life Sciences Building, 24 Tyndall Avenue, Bristol BS8 1TQ, UK

(iD) PCJD, 0000-0003-3116-7463; MR, 0000-0002-7254-837X

Teeth were an important innovation in vertebrate evolution but basic aspects of early dental evolution remain poorly understood. Teeth differ from other odontode organs, like scales, in their organized, sequential pattern of replacement. However, tooth replacement patterns also vary between the major groups of jawed vertebrates. Although tooth replacement in stem-osteichthyans and extant species has been intensively studied it has been difficult to resolve scenarios for the evolution of osteichthyan tooth replacement because of a dearth of evidence from living and fossil sarcopterygian fishes. Here we provide new anatomical data informing patterns of tooth replacement in the Devonian sarcopterygian fishes *Onychodus*, *Eusthenopteron* and *Tiktaalik* and the living coelacanth *Latimeria* based on microfocus- and synchrotron radiation-based X-ray microtomography. Early sarcopterygians generated replacement teeth on the jaw surface in a pattern similar to stem-osteichthyans, with damaged teeth resorbed and replacement teeth developed on the surface of the bone. However, resorption grades and development of replacement teeth vary spatially and temporally within the jaw. Particularly in *Onychodus*, where teeth were also shed through anterior rotation and resorption of bone at the base of the parasymphyseal tooth whorl, with new teeth added posteriorly. As tooth whorls are also present in more stem-osteichthyans, and statodont tooth whorls are present among acanthodians (putative stem-chondrichthyans), rotational replacement of the anterior dentition may be a stem-osteichthyan character. Our results suggest a more complex evolutionary history of tooth replacement.

## 1. Introduction

Teeth are a shared feature of all the jawed vertebrate (gnathostome) groups and were an important innovation in the radiation of jawed vertebrates [1,2]. Teeth are developmentally

and morphologically similar to dermal denticles [3–5] but differ in being localized to the jaw and exhibiting much greater temporal and spatial coordination in their patterning and replacement [3,4,6,7]. Tooth replacement enables worn or damaged teeth to be functionally replaced, and most living jawed vertebrates, except mammals, continuously replace their teeth throughout life [8,9]. However, the process of tooth replacement varies widely among gnathostome clades and tooth replacement characters are thus phylogenetically informative [1,2].

In the earliest jawed vertebrates, the placoderm fishes, there is evidence for organized successional teeth [10–12]. However, worn or damaged teeth and denticles are retained, with new teeth added in radial rows along one or more axes along the jaw [10,12]. Within Chondrichthyes, used teeth are not retained. Entire teeth, including the tooth base, are shed by linguolabial rotation of the tooth out of the jaw and replacement teeth are formed within a permanent dental lamina (a tooth generating epithelial organ) situated along the lingual jaw margin [6–8]. This contrasts with most fossil and extant osteichthyans where only the tooth crown is shed while the tooth base is resorbed to differing extents [7–9,13]. Replacement teeth develop predominantly on the surface of the jaw bone from either a permanent or non-permanent (transient) dental lamina positioned lingual of the functional tooth; so-called extraosseous, or extramedullary tooth replacement [9,14,15]. In amniotes and some teleosts, on the other hand, replacement teeth develop by down-growth of the epithelium into the medullary cavity of the jaw; so called intraosseous or intramedullary replacement [9,14]. Within sarcopterygians (or lobe-finned fishes, a major group of bony fishes that includes the tetrapods, lungfishes and coelacanths) teeth are replaced extraosseously in coelacanths and lissamphibians and intraosseously in amniotes [13,14,16]. Dipnoans, as early as the Late Devonian (360 Ma), had acquired a unique, non-replacing dentition [17].

As studies on dental development have mostly focused on extant osteichthyans and there is a paucity of evidence from early (putative stem-) osteichthyans, tooth replacement in basal sarcopterygians remains largely unknown [7–9,13,14,18]. Based on phylogenetic bracketing, basal sarcopterygians might be expected to replace their teeth extraosseously, with damaged teeth shed via basal resorption, as in modern coelacanths and stem-osteichthyans. This is supported by the distinctive replacement pits organized in an alternating pattern in osteolepiform and onychodontiform jaws [19,20]. However, it is also known that onychodontiforms and porolepiforms had parasymphyseal 'tooth whorls' with a potential rotational replacement mechanism, somewhat reminiscent of that in chondrichthyans [19,21,22]. This potentially indicates a greater diversity of sarcopterygian replacement patterns than is generally acknowledged.

Here, we use high-resolution microfocus- (XTM) and synchrotron radiation X-ray tomographic microscopy (SRXTM) to determine tooth replacement mechanisms in the Devonian sarcopterygians *Onychodus jandemarrai*, *Eusthenopteron foordi* and *Tiktaalik roseae*, and the extant coelacanth, *Latimeria chalumnae*. Our aim is to add anatomical data to the debate about ancestral modes of tooth replacement within the sarcopterygian fishes, analyse the mode and degree of tooth resorption and consider its implications for the evolutionary history of dental development and replacement in sarcopterygians.

## 2. Material and methods

Specimens were obtained from the Natural History Museum, London, United Kingdom (NHM UK), Naturhistoriska Riksmuseet, Stockholm, Sweden (NRM), Nunavet Fossil Vertebrate collection (NUFV), Canadian Museum of Nature, Ottawa, Ontario, Canada (NMC); tomographic data were obtained from the Muséum National d'Histoire Naturelle, Paris, France (MNHN). Scan data of the following specimens were used: a left maxilla, left dentary, left ectopterygoid and parasymphyseal tooth whorl of *Onychodus jandemarrai* (NHMUK PV P63570, Frasnian, Gogo Formation, Western Australia) [18]; an anterior right dentary fragment of *Eusthenopteron foordi* (NRM-PAL P.35; Frasnian, Escuminac Formation, Miguasha National Park, Canada) [20,23,24]; a left anterior dentary fragment of *Tiktaalik roseae* (NUFV666; Frasnian, Fram Formation, Nunavut territory, Canada); and a coronoid 2 of *Latimeria chalumnae* (MNHN-ZA-AC-2012-26, caught in Domoni, Comores) [25].

NHMUK PV P63570 and NRM-PAL P.35 were scanned at Naturalis Biodiversity Center, Leiden, The Netherlands, using a Zeiss Xradia 520 Versa micro-CT scanner. NHMUK PV P63570 was also scanned using synchrotron radiation X-ray tomographic microscopy (SRXTM) [26,27] at the TOMCAT beamline (X02DA), Swiss Light Source (SLS), Paul Scherrer Institut (PSI), Villigen, Switzerland [28]. Specimen NUFV666 was scanned at Carleton University, Ottawa, Ontario, Canada with a Skyscan 1173 micro-CT scan and MNHN-ZA-AC-2012-26 was scanned by [25] at the AST-RX platform of the Muséum National d'Histoire Naturelle, Paris, France, using a GE Sensing and Inspection Technologies phoenix X-ray v|tome|x L240-180 CT scanner. Scan parameters are reported in table 1. Radiographs were

reconstructed and subsequently analysed using Avizo 9.4.0/9.5.0. Volume renderings, segmentations and virtual sections were used to visualize the arrangement of the dentition and trace replacement pits, dentine remnants and resorption lines [7,12]. These tomographic data and 3D models are available at Dryad: https://dx.doi.org/10.5061/dryad.3nj1k8s following best practice guidelines for three-dimensional digital morphology data [29].

# 3. Results

## 3.1. Latimeria chalumnae

Coronoid two, an element of the inner dental arcade of the lower jaw (figure 1), contains one large tooth and several smaller teeth and oral denticles. The large tooth (often referred to as a 'fang', as followed here) [25] is positioned lingually from the smaller teeth. Oral denticles [4,5] are present across the labial and occlusal surface of the coronoid. Dentine, attachment bone and coronoid bone [25] can be differentiated based on differential absorption in the XTM data (figure 1b,d). Longitudinal ridges of the basal inner surface of the pulp cavity are considered to represent plies, infolds of dentine (figure 1c,d), characteristic for plicidentine of the simplexodont type by Meunier [25]. This structure does not fulfil the definition of plicidentine because the ridges are restricted to the implantation zone of the tooth and do not extend into the crown and follow the morphology of the external surface of the tooth [30]. The fang is attached to the lingual side of the coronoid by vascularized attachment bone in the socket [31]. The other teeth are ankylosed in similar fashion to the occlusal side of the coronoid. On the lingual side of the coronoid, there are multiple cup-shaped depressions on the occlusal surface (figure 1e). No resorption lines or vestigial dentine bodies, which might be interpretable as remnants of partially resorbed teeth, were observed under, or within, the bone of attachment of the fang or smaller teeth. The small depressions likely represent empty sockets of basally resorbed and shed teeth, referred to as 'replacement pits'.

## 3.2. Onychodus jandemarrai

The dentition of Onychodus consists of outer and inner dental arcades on the upper and lower jaws, organized in rows of fangs (or tusks after [19]), teeth and oral denticles and parasymphyseal tooth whorls in the lower jaw with fangs and denticles.

We examined elements of the outer dental arcade from the left maxilla (figure 2) and dentary (figure 3). They contain large teeth, organized in a lingual row along the axis of the jaw, with a row of smaller denticles present along the labial margin (figures 2a and 3a). The labial side of the maxilla and dentary is covered with granule-like denticles. Near the labial margin, denticles are more tooth-like (figures 2a,b,e,f and 3a–e). Dentine remnants, with distinct pulp cavities, of more tooth-like elements are present under the granule-like denticles along the labial side and are here interpreted to have been overgrown (figures 2f and 3d,e). Neither granule- or tooth-like denticles are organized in distinct rows or 'tooth-families'. The teeth are ankylosed to the dentary by attachment bone on the labial and ventral side of their bases (figures 2d,e and 3c). The labial denticles are fixed through attachment bone on the ventral side to the dentary. Both the row of teeth and the lateral denticle row contain a multitude of replacement pits (figures 2a and 3a). The replacement pits are interspersed in an alternating pattern along the row of fangs on the maxilla, anteriorly with one tooth and one empty socket, posteriorly with two teeth alternating with one empty socket (figure 2a). The dentary exhibits a more variable, almost alternating pattern (figure 3a). Replacement pits on the lateral denticle rows lack this pattern. Multiple layers of attachment bone, interspersed with resorption lines, remain under teeth, denticles and replacement pits (figures 2b and 3b). On their lingual side, the jaw elements are augmented with layers of bone interspersed by growth arrest lines (figure 4).

The inner dental arcade of the upper jaw, the ectopterygoid (figure 4), has one fang located lingually, close to its posterior margin. A row of smaller teeth occurs lingually, along the margin of the bone on the posterior half and the anterior half (figure 4a). The row is lined with less organized oral denticles; on the labial side these are not organized in a single row. The dentine base of the teeth is ankylosed to the ectopterygoid with attachment bone (figure 6e,f). Resorption lines are visible beneath teeth and oral denticles, (figure 4b–g). Pulp cavities of resorbed denticles are located under, and connected with pulp cavities of, standing denticle (figure 4f). On the posterior end of the ectopterygoid, vestigial dentine of resorbed denticles also occurs below denticles (figure 4b,c). The posterior half of the ectopterygoid contains lines of arrested growth along the lingual side, while the anterior half of this bone contains lines of arrested growth along both the lingual and the labial side (figure 4h,i).

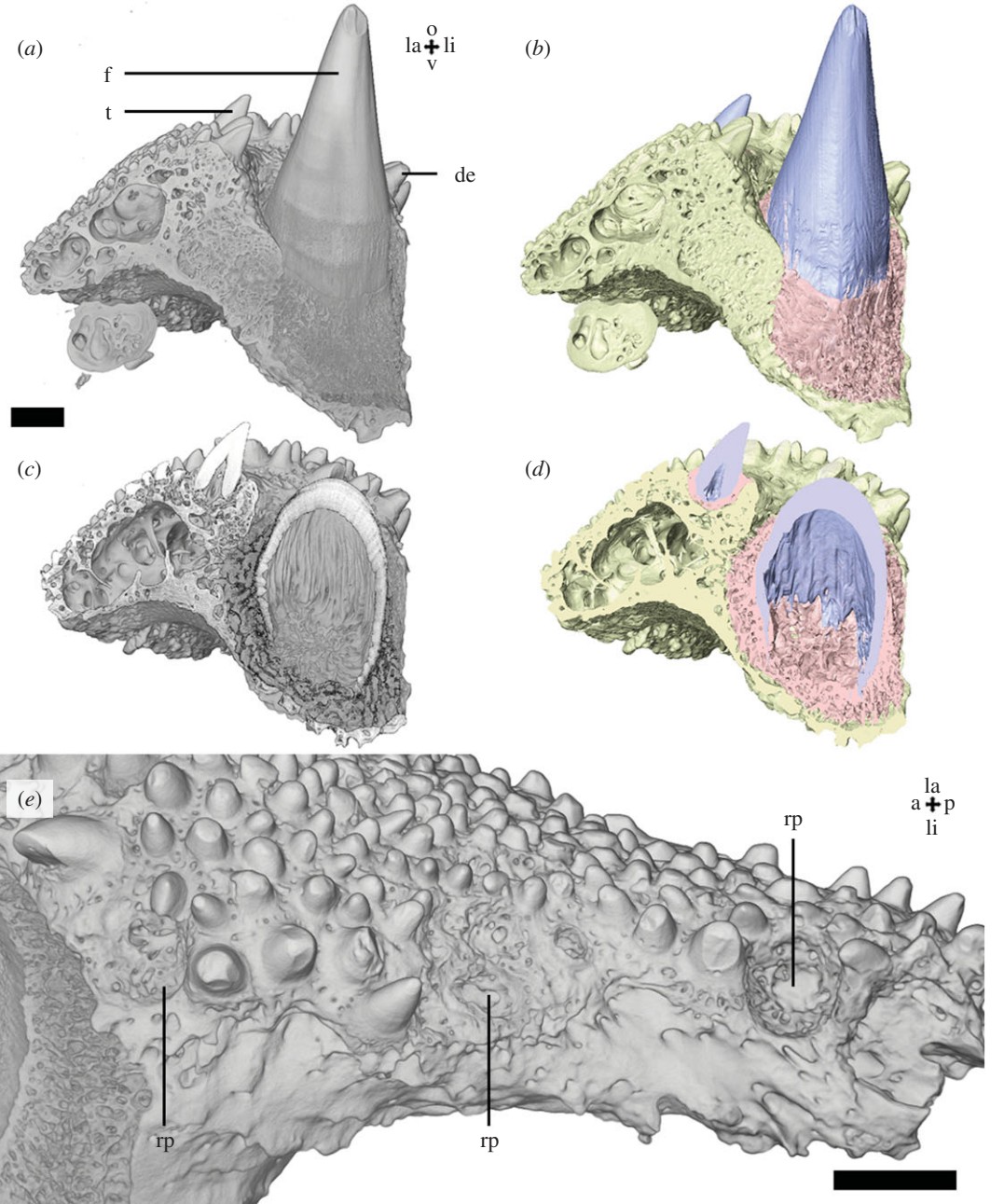

**Figure 1.** *Latimeria chalumnae* coronoid 2 (MNHN-ZA-AC-2012-26) micro-CT data, scan Latimeria. Anterior view (*a,b*) of the posterior half of the coronoid shows the position and organization of the dentition. Vertical sections (*c,d*) through the fang and a smaller tooth show the structure of the teeth and their bone of attachment. Colour segmentations (*b,d*) show the distinction between dentine, attachment bone and the coronoid. Lingual surface of the coronoid showing cup-shaped depression, replacement pits (e). Scalebar, 2 mm. a, anterior; de, denticle; f, fang; la, labial; li, lingual; o, occlusal; p, posterior; rp, replacement pit; t, tooth; v, ventral. Colours: blue = dentine, red = attachment bone, yellow = coronoid bone. Lingual surface of the coronoid showing cup-shaped depression, replacement pits (e).

The right parasymphyseal tooth whorl (figure 5) contains five large teeth, or fangs, and lateral rows of denticles on both sides of the fangs (figure 5*a*). The denticles are smaller and more numerous on the lateral (figure 5*d*), as opposed the medial side (figure 5*e*). The bases of fangs and denticles are ankylosed directly to the bone of the whorl (figure 5*b–e*). No dentine or tooth-like structures are present within the whorl bone or under the standing dentition. The bony base of the whorl does not extend beneath the anterior-most fang, and appears to have been resorbed, while the dentine base of the fang is intact (figure 5*c*). The bone does not contain growth arrest lines internally, however, growth lines are present along both sides of the whorl [19].

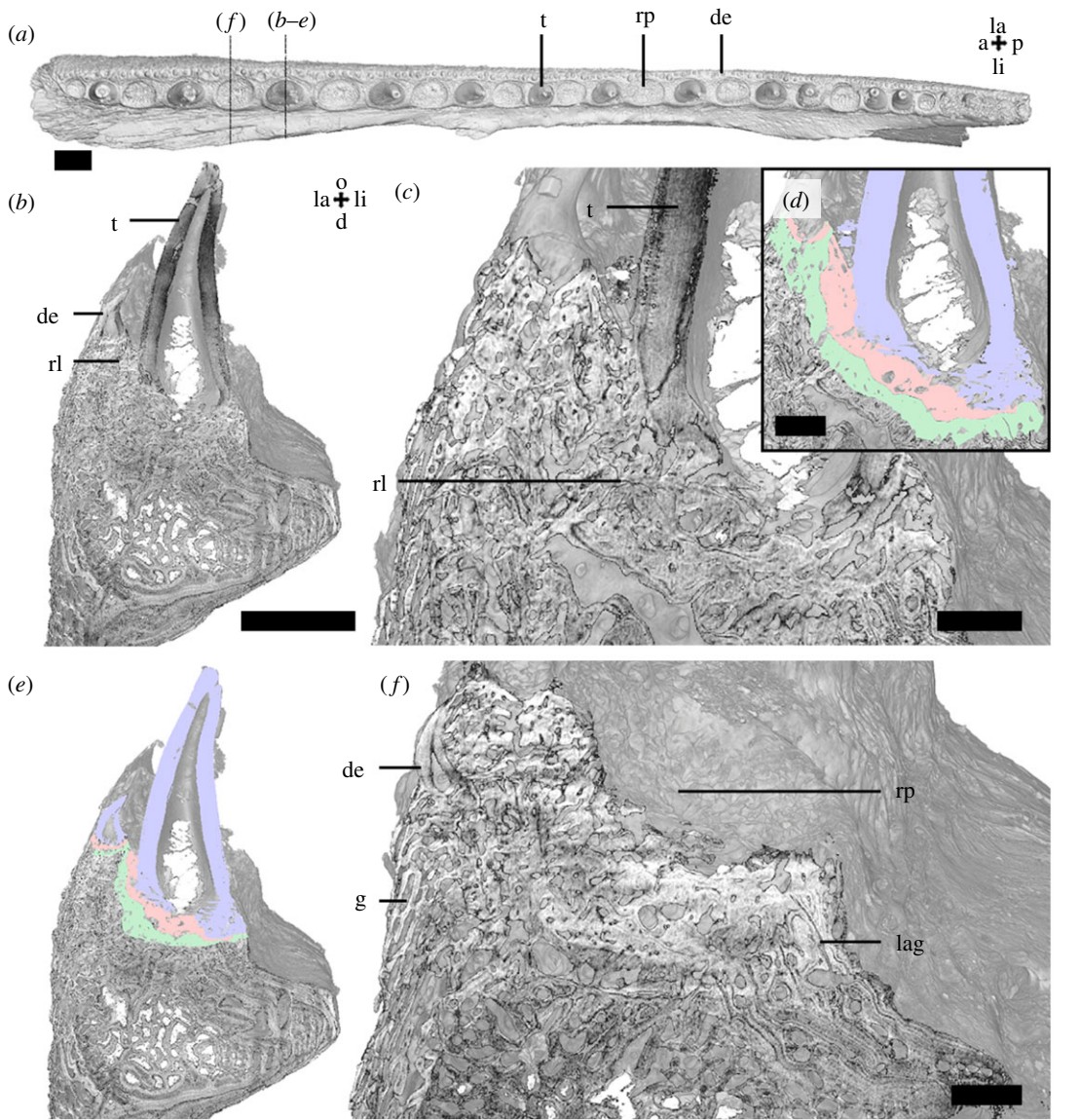

**Figure 2.** *Onychodus jandemarrai* maxilla (NHMUK PV P63570) micro-CT data, scans MDXR01a (*a*) and MDXR01d (*b–f*). Occlusal view (*a*) shows the position and organization of the dentition and replacement pits. Vertical sections in anterior view (*b–f*) through the third most distal fang and replacement pit show the structure of the dentition. Colour segmentations (*d,e*) show the distinction between dentine and attachment bone. Scalebar, 2 mm (*a,b,e*), 0.5 mm (*c,d,f*). a, anterior; d, dorsal; de, dentine; f, fang; g, granule; la, labial; lag, line of arrested growth; li, lingual; o, occlusal; p, posterior; rl, resorption line; rp, replacement pit; t, tooth. Colours: blue = dentine, green = attachment bone of resorbed tooth, red = attachment bone of standing tooth.

### 3.3. *Eusthenopteron foordi*

The anterior fragment of the right lower jaw (figure 6) contains the outer dental arcade with the dentary and the inner dental arcade with two coronoids. Tooth-like denticles with distinct pulp cavities are located along the labial margin of the dentary and in a tooth row lingually. The labial and ventral sides of the dentary are covered with granules lacking a pulp cavity.

The coronoids are separated by an intercoronoid fossa. The labial margin of the coronoids is lined with a single row of teeth. A pair of fangs occurs on the lingual side of every coronoid. One fang of each pair is incomplete (figure 6a). The ventral sides of the fangs are ankylosed by thin layers of attachment bone to the labial margin of the coronoid (figure 6c).

Polyplocodont plicidentine is typical for the fangs and smaller teeth. Replacement pits on the dentary and coronoid tooth rows do not follow an alternating pattern. Often two or more standing teeth or resorption pits occur adjacently. Vestigial plicidentine is observed under some replacement pits (figure 6e). Of the incomplete fangs, the bone of attachment and the plicidentine of the tooth

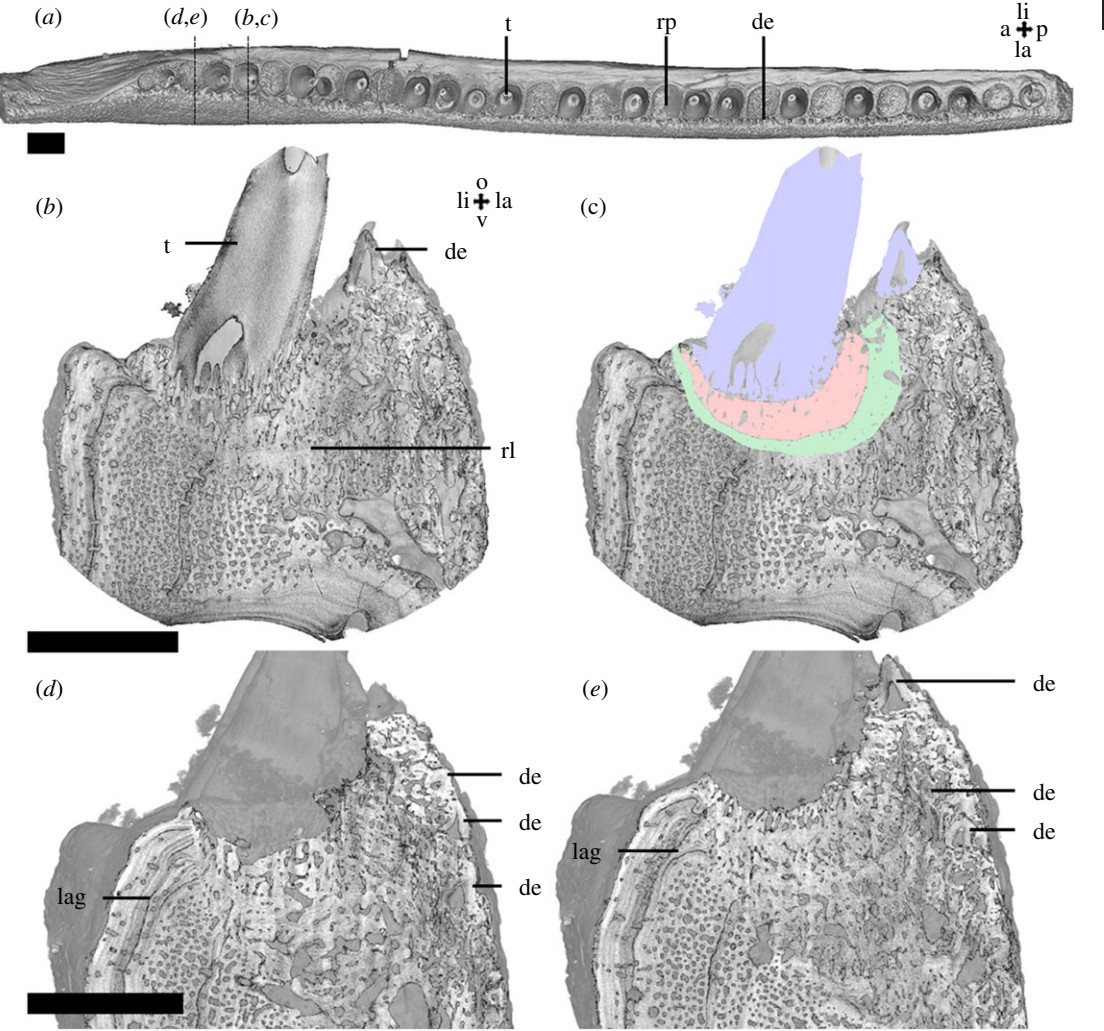

**Figure 3.** *Onychodus jandemarrai* dentary (NHMUK PV P63570) micro-CT data, scans MDXR02a (*a*) and MDXR02b (*b–e*). Occlusal view (*a*) shows the position and organization of the dentition and replacement pits. A vertical section in anterior view (*b*) through the third most anterior fang shows the structure of the fang, a labial tooth and their bone of attachment. A segmentation (*c*) shows interpretation of these structures. Scalebar, 2 mm. a, anterior; de, denticle; f, fang; la, labial; lag, line of arrested growth; li, lingual; o, occlusal; p, posterior; rl, resorption line; rp, replacement pit; t, smaller tooth; v, ventral. Colours: blue = dentine, green = bone of attachment of resorbed tooth, red = bone of attachment of standing tooth.

base remain, as under replacement pits of the other teeth. This reflects partial resorption of the teeth. However, no resorption lines or vestigial dentine were observed beneath the standing fangs (figure 6*b*).

## 3.4. Tiktaalik roseae

The fragment of the most anterior part of the left lower jaw contains a dentary (figure 7) and a coronoid with a fang pair and a lateral row of smaller teeth along the labial margin of the dentary, lined with oral denticles on the labial side. The labial and ventral sides of the dentary are covered with granules lacking pulp cavities. Near the symphysis, the dentary contains the dentine remnants of a smaller fang (figure 7*c*). On the coronoid, the more distal fang of the pair is missing, leaving a replacement pit and vestigial dentine of the tooth base (figure 7*c,d*). On the labial tooth row some teeth are also missing, leaving resorption pits (figure 7*a,b*). The pits are not arranged in a coherent pattern and no vestigial dentine or attachment bone is observed underneath (figure 7*a*). No resorption lines are observed under the fangs or the labial dentition. Plicidentine is typical for the teeth and fangs (figure 7*a,c,e*).

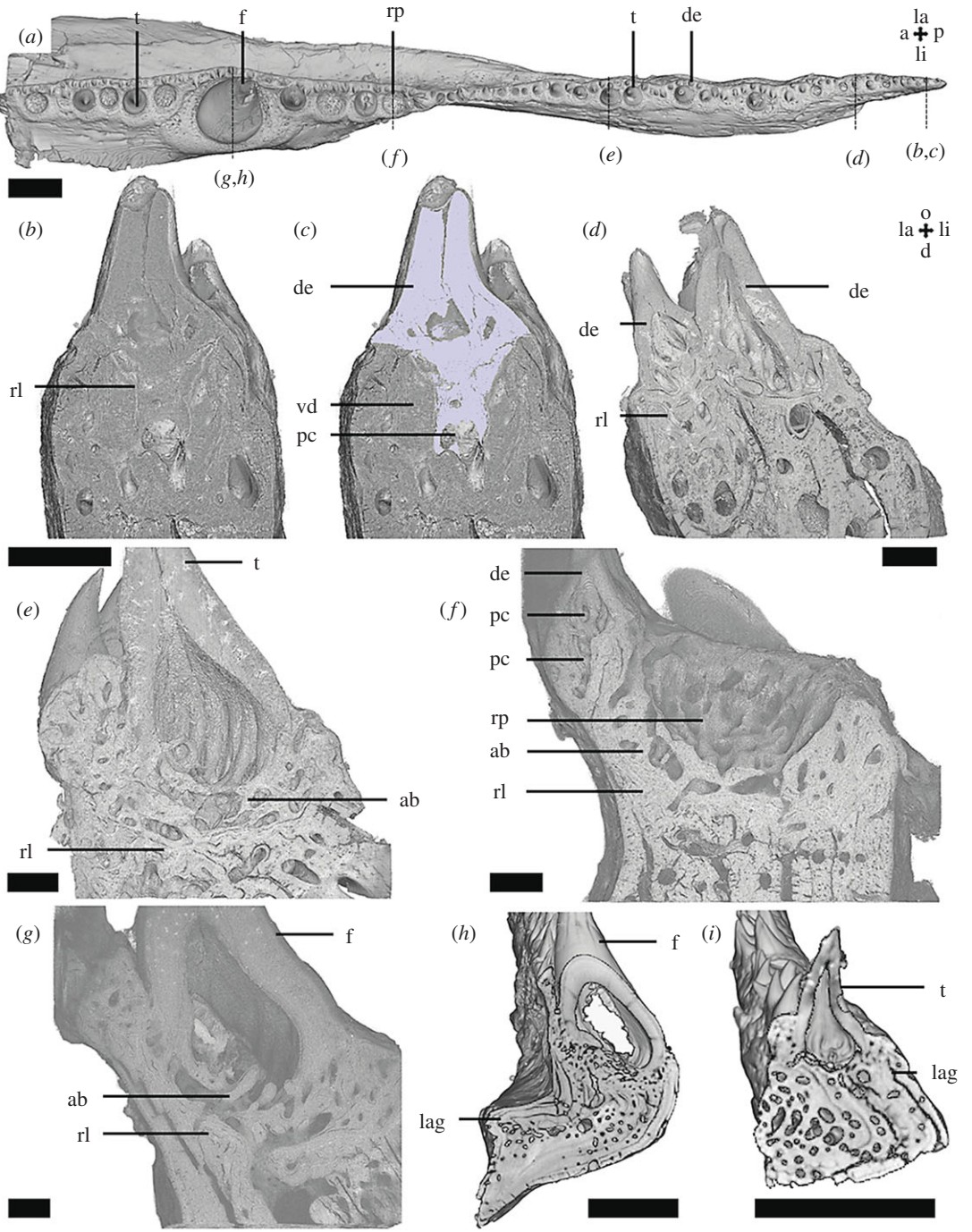

**Figure 4.** *Onychodus jandemarrai* (NHMUK PV P63570) Micro-CT and SRXTM data: scans MDXR03 (*a,h,i*), MR153a (*b,c*), MDXR03_2b (*d*) MDXR03_6c (*e*), MDXR03_10c (*f*) and MDXR03_16 (*g*). Occlusal view of the ectopterygoid (*a*) shows the position and organization of the dentition. Vertical sections in anterior view through the second most posterior denticle (*b,c*), a pair of denticles (*d*) and a tooth (*e,i*) on the posterior half and a replacement pit (*f*) and fang (*g,h*) on the anterior half of the ectopterygoid show the structure of the dentition and the underlying bone. Scalebar, 2 mm (*a,h,i*), 0.2 mm (*b–g*). a, anterior; ab, attachment bone; d, dorsal; de, denticle; f, fang; gl, growth line; la, labial; li, lingual; o, occlusal; p, posterior; pc, pulp cavity; rl, resorption line; rp, replacement pit; t, smaller tooth; vd, vestigial dentine. Colours: blue = dentine.

## 4. Discussion

Our survey of the structure of tooth attachment and replacement in sarcopterygian fishes reveals evidence of tooth shedding by basal resorption, with replacement teeth overgrowing resorbed earlier dental generations. Tooth replacement, based mainly on histological differences of teeth in

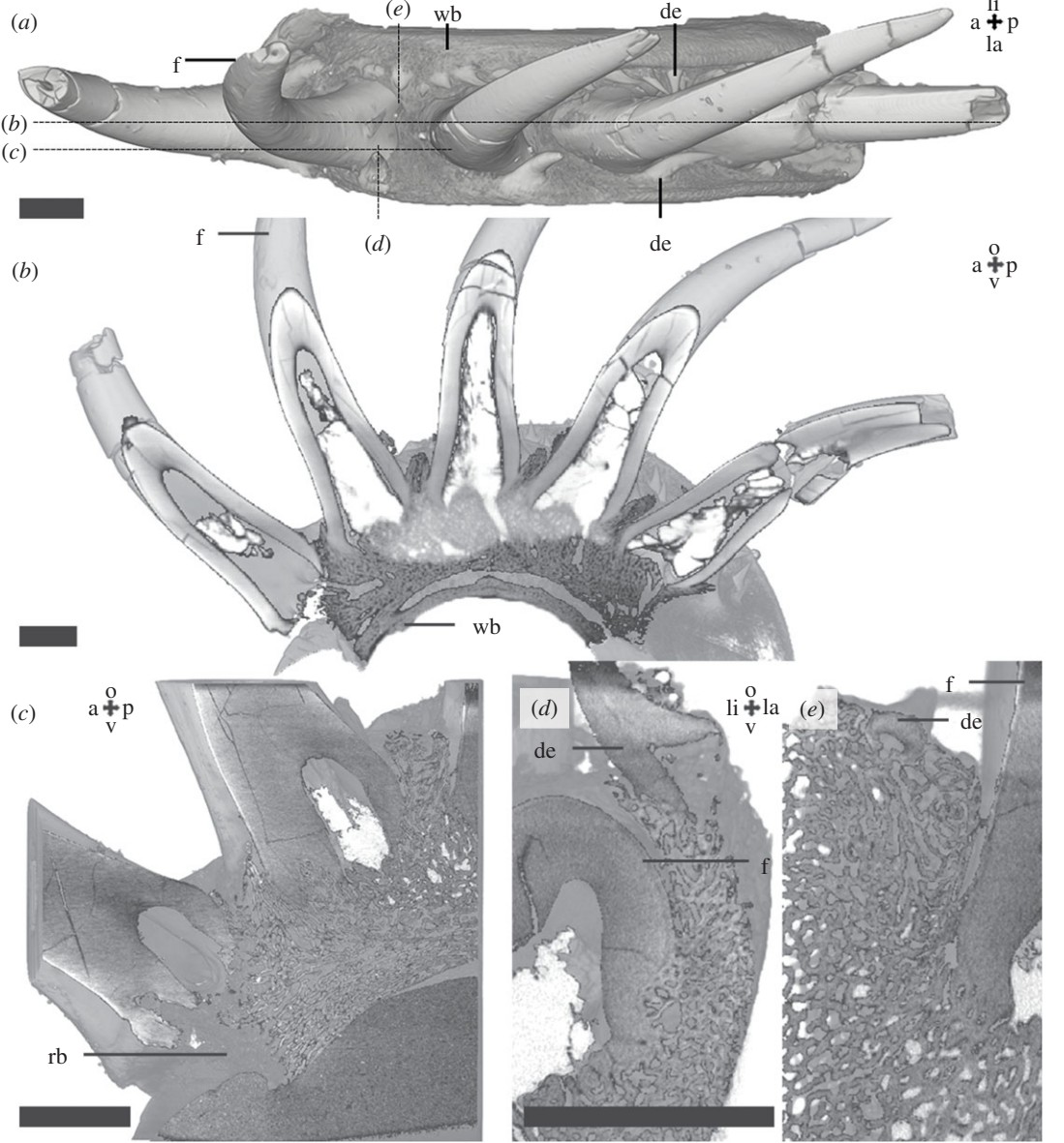

**Figure 5.** *Onychodus jandemarrai* parasymphyseal tooth whorl (NHMUK PV P63570) micro-CT data, scans XMR3a (*a,b*) and XMR3b (*c–e*). Occlusal view (*a*) shows the position and organization of the dentition. A longitudinal section (*b,c*) shows the structure of the dentition and the whorl bone. Vertical sections in anterior view (*d–e*) show the structure of the smaller teeth on the lateral rows. Scalebar, 2 mm. a, anterior; de, denticle; f, fang; la, labial; li, lingual; o, occlusal; p, posterior; rb, resorbed bone; v, ventral; wb, whorl bone.

sarcopterygians, has been described in detail [30,32–35]. However, reconstructing the developmental evolution and identifying the degree of resorption and the spatial and temporal development of replacement teeth using tomographic data demonstrate greater variation of tooth replacement than previously described.

In the outer dental arcade, the maxilla and dentary of *Onychodus*, we identify three types of overgrowing odontodes: dermal granule-like denticles, tooth-like denticles and teeth. Denticles on the labial side, are tooth-like, and show appositional growth and resorption. The presence of multiple layers of attachment bone under the denticle and tooth rows points to continuous basal resorption of teeth without vestigial dentine remaining. Replacement teeth formed either lingually, on-site, one-for-one, or by overgrowing their resorbed predecessors. However, tooth-like denticles are resorbed to varying extents leaving either only dentine remnants or distinct overgrown tooth-like denticles with pulp cavities on the labial side. The extent of denticle resorption varies and does not follow a distinct pattern. Replacement denticles tend to be added appositionally in a less organized way, especially

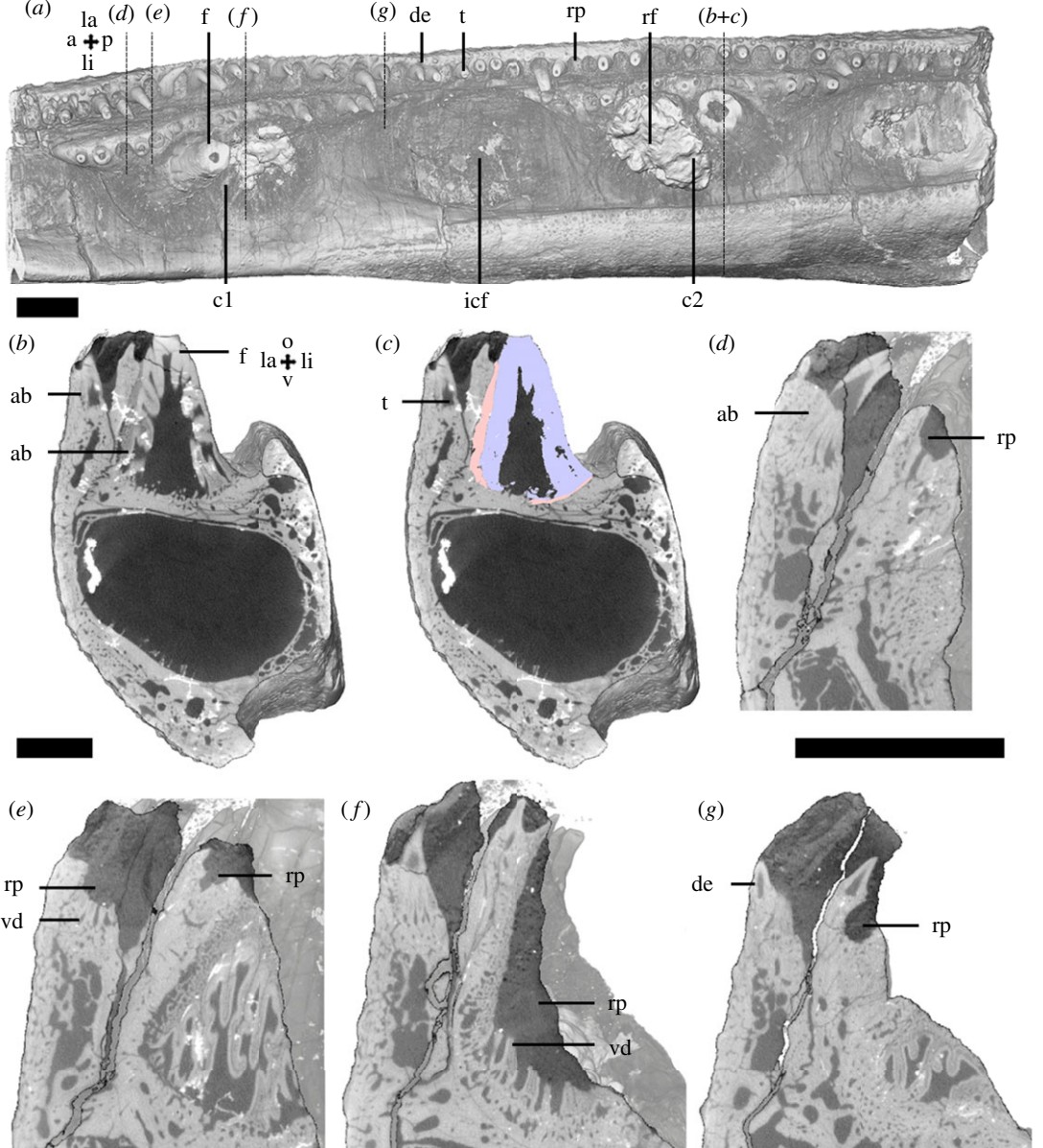

**Figure 6.** Dentary of *Eusthenopteron foordi* (NRM-PAL P.35) micro-CT data, scans MDXR05a and Efoordi1. Occlusal view (*a*) shows the position and organization of the dentition. Vertical sections in anterior view (*b*, *d–g*) through the dentary and the coronoid tooth rows show the structure of the dentition. A segmentation (*c*) of the distal fang of coronoid 2 shows the distinction between plicidentine and attachment bone. Scalebar, 2 mm. a, anterior; ab, attachment bone; c1, coronoid 1; c2, coronoid 2; de, dentine; f, fang; icf, intercoronoid fossa; la, labial; li, lingual; o, occlusal; p, posterior; rf, resorbed fang; rp, replacement pit; t, tooth; v, ventral; vd, vestigial dentine. Colours: blue = dentine, red = bone of attachment of standing tooth.

compared with the alternating replacement of teeth on the maxilla. Growth arrest lines indicate lateral growth of the jaw principally in a lingual direction to accommodate the increasing diameter of teeth in the tooth rows. Lateral growth of the bone and number of overgrown teeth were greatest around the growth centre of the jawbone.

In the inner dental arcade, the ectopterygoid, larger and smaller teeth were resorbed and replaced in similar fashion, leaving layers of attachment bone and resorption lines. Denticles on the labial margin were partly resorbed, leaving pulp cavities and vestigial dentine under overgrowing denticles in the posterior part of the ectopterygoid. Tooth arrangement is not as organized posteriorly as it is anteriorly in the ectopterygoid, or as in the outer dental arcade of the lower and upper jaw. Growth arrest lines in the anterior half of the ectopterygoid indicate growth in both labial and lingual directions whereas lingually-directed growth dominates posteriorly, similar to the dentary and maxilla.

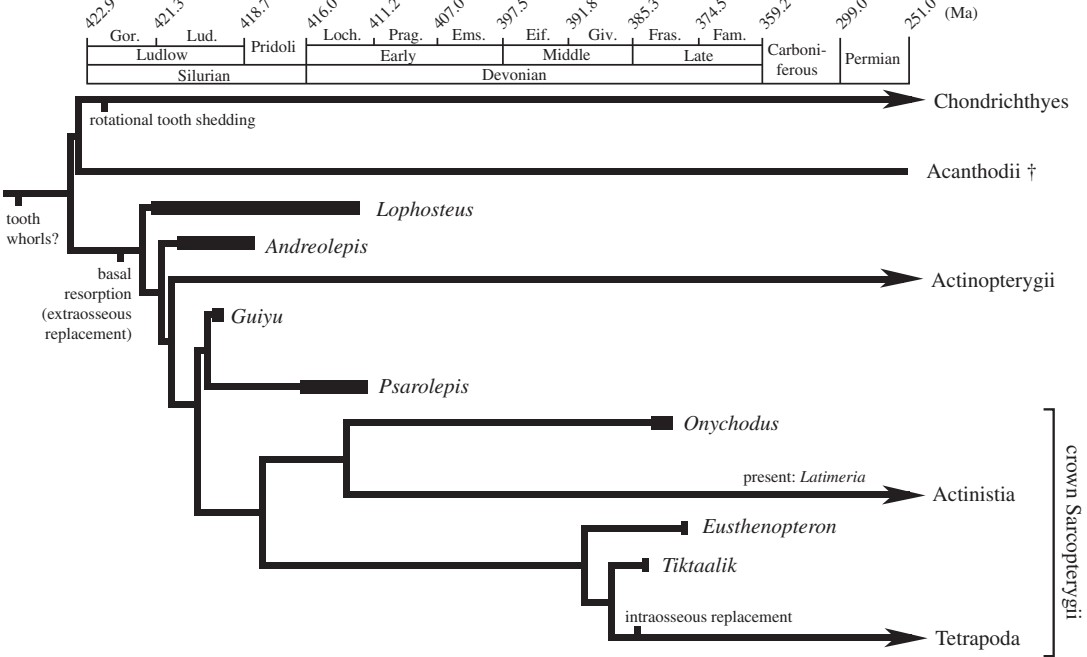

**Figure 7.** *Tiktaalik roseae* lower jaw fragment (NUFV666) micro-CT data. Occlusal view (*b*) shows the position and organization of the dentition, longitudinal (*a,c,d*) and vertical (*e*) sections in labial and anterior view through the dentary (*a*) and coronoid (*c–e*) dentition show the structure of the dentition. Scalebar, 2 mm. a, anterior; de, denticle; f, fang; g, granule; la, labial; li, lingual; o, occlusal; p, posterior; rp, replacement pit; t, tooth; v, ventral; vd, vestigial dentine.

**Figure 8.** Evolutionary hypothesis of tooth replacement in crown-gnathostomes. Time calibrated phylogeny follows [40].

**Table 1.** Parameters for X-ray and synchrotron tomographic scans. bin., camera binning; obj., objective.

| scan ID | species | specimen | material | machine | obj. | bin. | voxel size (µm) | voltage (kV) | voltage (keV) | exposure time (ms) |
|---|---|---|---|---|---|---|---|---|---|---|
| MDXR01a | O. jandemarrai | NHMUK PV P63570 | maxilla | Zeiss Xradia 520 Versa | 0.4× | 2 | 19.471 | 70 | | 4500 |
| MDXR01d | O. jandemarrai | NHMUK PV P63570 | maxilla | Zeiss Xradia 520 Versa | 0.4× | 2 | 4.6278 | 70 | | 4000 |
| MDXR02a | O. jandemarrai | NHMUK PV P63570 | dentary | Zeiss Xradia 520 Versa | 0.4× | 2 | 16.570 | 70 | | 9000 |
| MDXR02b | O. jandemarrai | NHMUK PV P63570 | dentary | Zeiss Xradia 520 Versa | 0.4× | 1 | 8.0476 | 70 | | 35 000 |
| MDXR03 | O. jandemarrai | NHMUK PV P63570 | ectopterygoid | Zeiss Xradia 520 Versa | 0.4× | 2 | 16.571 | 60 | | 15 000 |
| MDXR05a | E. foordi | NRM-PAL P.35 | dentary fragment | Zeiss Xradia 520 Versa | 0.4× | 2 | 13.834 | 80 | | 6000 |
| XMR3a | O. jandemarrai | NHMUK PV P63570 | tooth whorl | Zeiss Xradia 520 Versa | 0.4× | 2 | 33.117 | 60 | | 9000 |
| XMR3b | O. jandemarrai | NHMUK PV P63570 | tooth whorl | Zeiss Xradia 520 Versa | 0.4× | 2 | 8.3982 | 80 | | 8000 |
| Efoordi1 | E. foordi | NRM-PAL P.35 | dentary fragment | Zeiss Xradia 520 Versa | | | 11.189 | 80 | | |
| Latimeria | L. chalumnae | MNHN-ZA-AC-2012-26 | coronoid 2 | Phoenix L240-180 | | | 15 | 70 | | 333 |
| Tiktaalik | T. roseae | NUFV666 | dentary fragment | Skyscan 1173 | | 1 | 12.084 | 90 | | 1300 |
| MR153a | O. jandemarrai | NHMUK PV P63570 | ectopterygoid | TOMCAT beamline | 20× | 0 | 0.36 | | 23 | 999 |
| MDXR03_2b | O. jandemarrai | NHMUK PV P63570 | ectopterygoid | TOMCAT beamline | 10× | 0 | 0.65 | | 20 | 85 |
| MDXR03_6c | O. jandemarrai | NHMUK PV P63570 | ectopterygoid | TOMCAT beamline | 10× | 0 | 0.65 | | 20 | 85 |
| MDXR03_10c | O. jandemarrai | NHMUK PV P63570 | ectopterygoid | TOMCAT beamline | 10× | 0 | 0.65 | | 25 | 320 |
| MDXR03_16 | O. jandemarrai | NHMUK PV P63570 | ectopterygoid | TOMCAT beamline | 10× | 0 | 0.65 | | 32 | 360 |

*Eusthenopteron* and *Tiktaalik* also exhibit replacement pits and partially resorbed fangs on the dentary. Fang pairs were incomplete in both taxa, reflecting an alternating replacement of the fangs. Of the incomplete fangs, attachment bone and plicidentine of the tooth base remain, indicating the tooth crown had been shed by resorption of the tooth base. However, under the standing dentition, no resorption lines, vestigial dentine or layers of attachment bone were identified. This indicates a stepwise process with initial partial resorption and complete resorption of the tooth base together with the development of the replacement fang. For the dentary and coronoid tooth rows in *Eusthenopteron*, the grade of resorption varies between replacement pits, leaving vestigial dentine and attachment bone or neither. This suggests a shedding mechanism similar to that of the fangs. The different grade of resorption, compared to *Onychodus*, where plicidentine is absent, could be linked to the folded structure of the plicidentine dentition in *Eusthenopteron* and *Tiktaalik* and its inferred function to reinforce tooth attachment [25,36]. An outstanding difference is the thin bone of attachment under the fangs of *Eusthenopteron* compared to *Onychodus*, without plicidentine [30], which might also support the functional importance of the plicidentine in stabilizing the tooth attachment and connection. The absence of resorption and lines of arrested growth in *Eusthenopteron*, *Latimeria* or *Tiktaalik* might be linked to relatively (low) imaging resolution. In the case of *Eusthenopteron* and *Tiktaalik*, higher X-ray energy resulted in lower contrast limiting the visualization of subtle density differences. Nevertheless, dentine tissues of partially resorbed teeth are visible and their absence in some instances seems to be largely a result of complete resorption rather than an imaging artefact.

There is a wealth of descriptions of resorption and dental development in sarcopterygians based on thin sections, e.g. [30,32–34]. However, there have been no reconstructions of tooth order after resorption and the replacement mechanisms enabled through our novel tomographic data. Although we find a basic conservation of extraosseous replacement within early sarcopterygians our results also reveal previously unrecognized diversity in replacement processes both between tooth types, and between teeth and denticles. In the lingual maxillary and dentary rows of *Eusthenopteron* and *Tiktaalik*, and the anterior tooth row (ectopterygoid) of *Onychodus*, teeth were replaced in an almost alternating pattern similar to other onychodontiforms and osteolepiforms [19,20]. This alternating replacement within distinct rows, or fang pairs (in the case of *Eusthenopteron* and *Tiktaalik*), was traditionally used to infer the presence of a dental lamina [5,37]. However, formation of an organized pattern without a dental lamina has also been demonstrated [9,38] and the inference of developmental structures like a dental lamina based on fossil representatives is problematic. As with extant *Latimeria*, replacement teeth in these Devonian sarcopterygians likely formed in the same position as their resorbed precursors, from epithelium restricted to the replacement pit [13]. The development of replacement teeth with fully resorbed precursors contrasts markedly with how new denticles tended to overgrow older denticles in a 'gap-filling' arrangement on the coronoid of *Latimeria* and along the labial margin of the *Onychodus* jaw elements [5,7]. Our results thus highlight how the patterning arrangements of teeth and denticles vary markedly both spatially and during growth, as well as suggesting marked differences in regenerative processes between teeth and denticles.

Even more disparate replacement mechanisms were identified for the parasymphyseal tooth whorl of *Onychodus*. Evolutionary scenarios for tooth replacement in early vertebrates have tended to simplify the picture by attributing replacement to a uniform and singular process. However, *Onychodus* demonstrates that replacement of the parasymphyseal teeth differs fundamentally from the replacement of the posterior dentition. The most distal tooth of the parasymphyseal tooth whorl shows evidence of bone resorption at the base, while the dentine of the tooth base remained intact. Teeth are added proximally and progress anteriorly through rotation of the whorl, eventually leading to shedding of the anterior tooth [19]. In other *Onychodus* specimens, unattached teeth on the distal end of the whorl, and growth lines along the lateral and medial side of the tooth whorl, indicate that rotational replacement probably operated by proximal addition of teeth and bone to the whorl [19].

The presence of parasymphyseal tooth whorls among onychodontiforms, porolepiforms and putative stem-sarcopterygians like *Guiyu* and *Psarolepis* (figure 8) [19,21,39–41] argues for their plesiomorphy among sarcopterygians and possibly osteichthyans. Although tooth whorls have not been associated with *Andreolepis hedei* [22], tooth whorls with a bony base are present in acanthodians, putative stem-chondrichthyans (figure 8) [7,39,42]. Tomographic data from acanthodian tooth whorls indicate that they develop through proximal addition of teeth and bone as in *Onychodus* [39,43]. However, acanthodian tooth whorls are statodont (lack a shedding mechanism and teeth are retained), like placoderm tooth rows [10,12], which suggests a separate evolution of the shedding mode of Chondrichthyes without resorption, different to the osteichthyan condition of replacement with

resorption prior to shedding (figure 8). Rotational replacement might represent the ancestral condition for Osteichthyes or even crown gnathostomes. More sampling, combined with detailed examination of tooth formation and replacement and phylogenetic analysis, will help resolve questions about the homology of dental organs and evolutionary pattern of tooth replacement in crown gnathostomes [21,22,40,41,44–48].

## 5. Conclusion

Micro-CT and SRXTM allow us to add substantial data to reconstruct tooth resorption and tooth addition and infer the mode of tooth replacement at the base of crown-Sarcopterygii. Resorption and extraosseous replacement of the marginal dentition are present in all studied taxa. This supports the hypothesis that extraosseous tooth replacement is an ancestral condition for crown-Sarcopterygii and crown-Osteichthyes. Variation and differentiation of dental morphogenesis along the marginal jaw is evident, with resorption grades and development of replacement teeth varying spatially and temporally. Symphyseal dentition, exemplified by the parasymphyseal tooth whorl of *Onychodus*, exhibits a fundamentally distinct replacement pattern. Our results thus support the possibility of a more complex evolution of tooth replacement. Rotational replacement of the anterior dentition via a tooth whorl mechanism is likely to be an ancestral osteichthyan or crown gnathostome trait. More data on the variation of tooth formation and replacement among stem and crown osteichthyans, as well as stem-chondrichthyans, is needed to reconstruct dental developmental evolution.

Data accessibility. Scan and surface files are archived on the Dryad Digital Repository: https://dx.doi.org/10.5061/dryad.3nj1k8s.

Authors' contributions. M.R. and M.D. designed the study. M.D., M.R. and A.M.C.C. scanned the specimens. M.D. compared and described the material, processed and analysed the CT data, interpreted the results, prepared all the figures and the first draft. M.D., A.M.C.C., P.C.J.D. and M.R. wrote the paper and gave final approval for publication.

Competing interests. We declare we have no competing interests.

Funding. This work was supported by NWO Vidi grant no. 864.14.009 to M.R. and NERC grant no. NE/G016623/1 to P.C.J.D. We acknowledge the Paul Scherrer Institut, Villigen, Switzerland for provision of synchrotron radiation beamtime at the TOMCAT beamline X02DA of the SLS. Internal funding from the Naturalis Biodiversity Center supported travel costs, scanning and access to computing infrastructure.

Acknowledgements. Bertie-Joan van Heuven and Rob Langelaan scanned specimens at Naturalis and Fred Gaidies, Hillary Maddin, Kieran Shepherd provided access to and scanned the *Tiktaalik* specimen. Neil Shubin and Ted Daeschler provided information and helped getting access to the *Tiktaalik* specimen. Zerina Johanson (NHM UK) and Jonas Hagström (NRM) provided *Onychodus* and *Eusthenopteron* material respectively. We thank Federica Marone (PSI) for technical assistance with scanning at the TOMCAT beamline and Gaël Clément (MNHN) for sharing *Latimeria* scan data. We thank Benedict King (Naturalis) for fruitful discussions.

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
