## [Reviewer comments · Royal Society Open Science]

Review History

RSOS-191173.R0 (Original submission)

Review form: Reviewer 1

Is the manuscript scientifically sound in its present form?

Yes

Are the interpretations and conclusions justified by the results?

Yes

Is the language acceptable?

Yes

Do you have any ethical concerns with this paper?

No

Have you any concerns about statistical analyses in this paper?

No

Recommendation?

Major revision is needed (please make suggestions in comments)

Comments to the Author(s)

You neglect all old literature, and cite Peyer 1968 (the whole Odontology is by Peyer, translated and edited by Zangerl from a German manuscript of Peyer) wrongly in the bibliography: pp. 80-110, which deal with actinopterygians not sarcopterygians; sarcopterygians (in Peyer: Choanichthyes) can be found on pp. 114 ff. Therein Bystrow (1938) is cited and figured for tooth re-placement in early tetrapods. Bystrow (1939, 1942) described tooth replacement in piscine sarcopterygians. If the authors dislike Bystrow's idealized figures, they can check Schultze (1969).

There is no proof that plicidentine enhances tooth attachment. The term plicidentine makes only sense defined as folded orthodontine independent from the external tooth surface (Schultze 1969, p. 66 contra Meunier). Meunier described interaction between attachment bone and dentine as plicidentine. Furrows on the outside of the tooth base invaded by attachment bone enhance of course the stability of a tooth attachment, nevertheless that is not plicidentine. In that sense *Onychodus* has vertical folds at the base, into which attachment bone reaches like an electric plug into an electric outlet (Schultze 1968).

Bystrow (1938, 1939, 1942, *Ark. Zool.* 19, 20, 23) and Schultze (1969, *Palaeontogr. Ital.* 65) described resorption extensively. In addition there is a lot of literature on Zahnreihen postulating resorption (see e.g. de Riques & Bolt 1983, *JVP* 3).

Jessen (1968, *Ark. Zool. sr.* 2, 18) argued for difference of the symphyseal whorl between onychodonts and porolepiforms. The sister group of porolepiforms, the dipnoans, don't possess a tooth whorl, the actinistians, the sister of onychodonts, neither. Only a detailed mapping of tooth whorls on a detailed cladogram can justify an argument of the appearance of the first tooth whorl.

Review form: Reviewer 2 (Moya Meredith Smith)

Is the manuscript scientifically sound in its present form?

Yes

Are the interpretations and conclusions justified by the results?

Yes

Is the language acceptable?

Yes

Do you have any ethical concerns with this paper?

No

Have you any concerns about statistical analyses in this paper?

No

Recommendation?

Accept with minor revision (please list in comments)

Comments to the Author(s)

I thought 2 refs should have been added in the discussion of placoderm teeth in the introduction as they were the first to suggest placoderms had teeth in a creation order that was sequential along the row. Also upper tooth plates had radial rows that could have led to rotational succession.

Placoderm fishes, pharyngeal denticles and the vertebrate dentition.

Separate evolutionary origins of teeth from evidence in fossil jawed vertebrates.

2003. Smith Moya, M. and Johanson, Z. *Science*, 299, 1235-1236.

2003. Johanson, Z. and Smith Moya M. J. *Morphology*, 257, 289-307.

These as especially the paper reviews all conditions of how teeth are added in succession in stem groups for osteichthyan fish.

in the discussion too much was made of the tooth whorl in *Onychodus*, Lines 268 – 276- too much focus on a few examples of tooth whorls forming close to the symphysis and suggesting affinity with those of chondrichthyans, maybe independent of those in chondrichthyans. A morphotype of tooth sets that could occur many times.

Conclusions – that Rotational replacement of the anterior dentition via a tooth whorl mechanism is likely to be an ancestral osteichthyan or crown gnathostome trait.

Aim is to reconstruct dental developmental evolution, that was the aim of previous papers but this adds substantial data to this through the new techniques of segmentation and reconstruction of tooth order after a resorptive event, that is well done.

Decision letter (RSOS-191173.R0)

23-Sep-2019

Dear Dr Rücklin,

The editors assigned to your paper ("Tooth replacement in basal sarcopterygians") have now received comments from reviewers. We would like you to revise your paper in accordance with the referee and Associate Editor suggestions which can be found below (not including confidential reports to the Editor). Please note this decision does not guarantee eventual acceptance.

Please submit a copy of your revised paper before 16-Oct-2019. Please note that the revision deadline will expire at 00.00am on this date. If we do not hear from you within this time then it will be assumed that the paper has been withdrawn. In exceptional circumstances, extensions may be possible if agreed with the Editorial Office in advance. We do not allow multiple rounds of revision so we urge you to make every effort to fully address all of the comments at this stage. If deemed necessary by the Editors, your manuscript will be sent back to one or more of the original reviewers for assessment. If the original reviewers are not available, we may invite new reviewers.

- Data accessibility

If you wish to submit your supporting data or code to Dryad (<http://datadryad.org/>), or modify your current submission to dryad, please use the following link:
<http://datadryad.org/submit?journalID=RSOS&manu=RSOS-191173>

- Competing interests

- Authors' contributions

- Acknowledgements

- Funding statement

on behalf of Dr Michael Tobler (Associate Editor) and Kevin Padian (Subject Editor)
 openscience@royalsociety.org

Associate Editor's comments (Dr Michael Tobler):

Associate Editor: 1

Comments to the Author:

We received feedback from two reviewers that had an overall positive impression of the manuscript. Both reviewers felt that the authors need to better consider previously published work on this topic and clearly explain the novel aspects of their study.

Editor comments:

The revisions are straightforward but they should be completed before acceptance of the manuscript. Please address all the reviewers' concerns, and thanks for your submission.

Comments to Author:

Reviewers' Comments to Author:

Reviewer: 1

Comments to the Author(s)

You neglect all old literature, and cite Peyer 1968 (the whole Odontology is by Peyer, translated and edited by Zangerl from a German manuscript of Peyer) wrongly in the bibliography: pp. 80-110, which deal with actinopterygians not sarcopterygians; sarcopterygians (in Peyer: Choanichthyes) can be found on pp. 114 ff. Therein Bystrow (1938) is cited and figured for tooth re-placement in early tetrapods. Bystrow (1939, 1942) described tooth replacement in piscine sarcopterygians. If the authors dislike Bystrow's idealized figures, they can check Schultze (1969).

There is no proof that plicidentine enhances tooth attachment. The term plicidentine makes only sense defined as folded orthodontine independent from the external tooth surface (Schultze 1969, p. 66 contra Meunier). Meunier described interaction between attachment bone and dentine as plicidentine. Furrows on the outside of the tooth base invaded by attachment bone enhance of course the stability of a tooth attachment, nevertheless that is not plicidentine. In that sense Onychodus has vertical folds at the base, in to which attachment bone reaches like an electric plug into an electric outlet (Schultze 1968).

Bystrow (1938, 1939, 1942, Ark. Zool. 19, 20, 23) and Schultze (1969, Palaeontogr. Ital. 65) described resorption extensively. In addition there is a lot of literature on Zahnreihen postulating resorption (see e.g. de Riqles & Bolt 1983, JVP 3).

Jessen (1968, Ark. Zool. sr. 2, 18) argued for difference of the symphyial whorl between onychodonts and porolepiforms. The sister group of porolepiforms, the dipnoans, don't possess a tooth whorl, the actinistians, the sister of onychodonts, neither. Only a detailed mapping of tooth whorls on a detailed cladogram can justify an argument of the appearance of the first tooth whorl.

Reviewer: 2

Comments to the Author(s)

I thought 2 refs should have been added in the discussion of placoderm teeth in the introduction as they were the first to suggest placoderms had teeth in a creation order that was sequential along the row. Also upper tooth plates had radial rows that could have led to rotational succession.

Placoderm fishes, pharyngeal denticles and the vertebrate dentition.

Separate evolutionary origins of teeth from evidence in fossil jawed vertebrates.

2003. Smith Moya, M. and Johanson, Z. *Science*, 299, 1235-1236.

2003. Johanson, Z. and Smith Moya M. J. *Morphology*, 257, 289-307.

These as especially the paper reviews all conditions of how teeth are added in succession in stem groups for osteichthyan fish.

in the discussion too much was made of the tooth whorl in *Onychodus*, Lines 268 – 276- too much focus on a few examples of tooth whorls forming close to the symphysis and suggesting affinity with those of chondrichthyans, maybe independent of those in chondrichthyans. A morphotype of tooth sets that could occur many times.

Conclusions – that Rotational replacement of the anterior dentition via a tooth whorl mechanism is likely to be an ancestral osteichthyan or crown gnathostome trait.

Aim is to reconstruct dental developmental evolution, that was the aim of previous papers but this adds substantial data to this through the new techniques of segmentation and reconstruction of tooth order after a resorptive event, that is well done.

Author's Response to Decision Letter for (RSOS-191173.R0)

See Appendix A.

Decision letter (RSOS-191173.R1)

22-Oct-2019

Dear Dr Rücklin,

I am pleased to inform you that your manuscript entitled "Tooth replacement in early sarcopterygians" is now accepted for publication in Royal Society Open Science.

Kind regards,
Lianne Parkhouse
Editorial Coordinator
Royal Society Open Science
openscience@royalsociety.org

on behalf of Dr Michael Tobler (Associate Editor) and Kevin Padian (Subject Editor)
openscience@royalsociety.org

Appendix A

15 of October 2019

Dear Editors,

Re: Manuscript ID RSOS-191173

Please find enclosed a revision of our manuscript entitled 'Tooth replacement in basal sarcopterygians'. We have revised the manuscript in light of the comments raised by the referees and editors. In particular, we have cited more previously published work and explained more the novel aspects of our work.

Below we provide a point-for-point response indicating the changes that we have made, and an explanation where we have chosen not to follow the referees' advice.

We believe that we have addressed the points raised by the two reviewers in the revision of our manuscript.

We would be grateful if you could consider its case for publication.

Yours sincerely,

Martin Rücklin, Mark Doeland, Aidan Couzens and Philip Donoghue

Associate Editor's comments (Dr Michael Tobler):

Associate Editor: 1

Comments to the Author:

We received feedback from two reviewers that had an overall positive impression of the manuscript. Both reviewers felt that the authors need to better consider previously published work on this topic and clearly explain the novel aspects of their study.

We include more previously published work and emphasize the novel aspects of our study in more detail.

Editor comments:

The revisions are straightforward but they should be completed before acceptance of the manuscript. Please address all the reviewers' concerns, and thanks for your submission.

We believe that we have addressed all reviewer comments.

Comments to Author:

Reviewers' Comments to Author:

Reviewer: 1

Comments to the Author(s)

You neglect all old literature, and cite Peyer 1968 (the whole Odontology is by Peyer, translated and edited by Zangerl from a German manuscript of Peyer) wrongly in the bibliography: pp. 80-110, which

deal with actinopterygians not sarcopterygians; sarcopterygians (in Peyer: Choanichthyes) can be found on pp. 114 ff.

We referred to the part dealing with plicidentine, see also Meunier et al. 2015. We correct the citation. We included already some "old literature" and include more on sarcopterygian dentitions as mentioned by the reviewer.

Therein Bystrow (1938) is cited and figured for tooth replacement in early tetrapods. Bystrow (1939, 1942) described tooth replacement in piscine sarcopterygians. If the authors dislike Bystrows idealized figures, they can check Schultze (1969).

We include Bystrow 1938, 1939, 1942, Schultze 1969 in the discussion as description of the condition in *Eusthenopteron*.

There is no proof that plicidentine enhances tooth attachment.

We agree and therefore we refer to the inferred function of the structure in our manuscript.

The term plicidentine makes only sense defined as folded orthodontine independent from the external tooth surface (Schultze 1969, p. 66 contra Meunier). Meunier described interaction between attachment bone and den-tine as plicidentine. Furrows on the outside of the tooth base invaded by attachment bone enhance of course the stability of a tooth attachment, nevertheless that is not plicidentine. In that sense *Onychodus* has vertical folds at the base, in to which attachment bone reaches like an electric plug into an electric outlet (Schultze 1968).

We add a brief discussion on the terminology to clarify that there is disagreement on the use of the term plicidentine especially for *Latimeria*. We don't describe plicidentine in *Onychodus* and agree with the definition of the reviewer.

Bystrow (1938, 1939, 1942, Ark. Zool. 19, 20, 23) and Schultze (1969, Palaeontogr. Ital. 65) described resorption extensively. In addition there is a lot of literature on Zahnreihen postulating resorption (see e.g. de Ricqlès & Bolt 1983, JVP 3).

We add the references and agree that there is resorption mentioned before, we did not claim to be the first ones recognizing this mechanism, but it was not studied in detail in the different rows in three dimensions. We did not include De Ricqlès & Bolt (1983). This work focusses on a specialized dentition in captorhinids with a derived form of postulated tooth replacement, as more recently described by LeBlanc & Reisz (2015), and we therefore consider this less relevant to our study.

Jessen (1968, Ark. Zool. sr. 2, 18) argued for difference of the symphyseal whorl between onychodonts and porolepiforms. The sister group of porolepiforms, the dipnoans, don't possess a tooth whorl, the actinistians, the sister of onychodonts, neither. Only a detailed mapping of tooth whorls on a detailed

cladogram can justify an argument of the appearance of the first tooth whorl.

The aim of our study is to describe the data and start a discussion, an extensive phylogenetic discussion using an ancestral state reconstruction is beyond the scope of this study. This will be the topic of a following up manuscript including sarcopterygians, osteichthyans, and chondrichthyans to identify the ancestral state of tooth whorls in crown gnathostomes.

Reviewer: 2

Comments to the Author(s)

I thought 2 refs should have been added in the discussion of placoderm teeth in the introduction as they were the first to suggest placoderms had teeth in a creation order that was sequential along the row. Also upper tooth plates had radial rows that could have led to rotational succession.

Placoderm fishes, pharyngeal denticles and the vertebrate dentition.

Separate evolutionary origins of teeth from evidence in fossil jawed vertebrates.

2003. Smith Moya, M. and Johanson, Z. *Science*, 299, 1235-1236.

2003. Johanson, Z. and Smith Moya M. *J. Morphology*, 257, 289-307.

These as especially the paper reviews all conditions of how teeth are added in succession in stem groups for osteichthyan fish.

We include Smith and Johanson 2003 in the introduction to complete the list of manuscripts cited, but the placoderm condition is not the main emphasis of the manuscript and only important for the introduction. The upper jaw condition does not show a tooth whorl and the inferred evolution of a rotational succession is not evident.

The denticles on the postbranchial wall of placoderms (Johanson and Smith 2003) were discussed in Rücklin et al. 2012 and identified as "...simple focal developments of continuous sheets of spongy bone, added episodically to the growing margin of the postbranchial wall". Instead we add the review paper of the same authors Johanson and Smith 2005.

in the discussion too much was made of the tooth whorl in *Onychodus*, Lines 268 – 276- too much focus on a few examples of tooth whorls forming close to the symphysis and suggesting affinity with those of chondrichthyans, maybe independent of those in chondrichthyans. A morphotype of tooth sets that could occur many times.

We shorten and rephrase this part accordingly.

Conclusions – that Rotational replacement of the anterior dentition via a tooth whorl mechanism is likely to be an ancestral osteichthyan or crown gnathostome trait.

Aim is to reconstruct dental developmental evolution, that was the aim of previous papers but this adds substantial data to this through the new techniques of segmentation and reconstruction of tooth order after a resorptive event, that is well done.

We simplify the conclusions and emphasize more the novel aspects as suggested by the reviewer.